# The Impact of IL-6 and IL-10 Gene Polymorphisms in Diffuse Large B-Cell Lymphoma Risk and Overall Survival in an Arab Population: A Case-Control Study

**DOI:** 10.3390/cancers12020382

**Published:** 2020-02-07

**Authors:** Sohaib M. Al-Khatib, Nour Abdo, Laith N. AL-Eitan, Abdel-Hameed Al-Mistarehi, Deeb Jamil Zahran, Tariq Zuheir Kewan

**Affiliations:** 1Department of Pathology and Laboratory Medicine, Faculty of Medicine, Jordan University of Science and Technology, Irbid 22110, Jordan; 2Department of Public Health, Faculty of Medicine, Jordan University of Science and Technology, Irbid 22110, Jordan; nmabdo@just.edu.jo; 3Department of Biotechnology and Genetic Engineering, Faculty of Science and Arts, Jordan University of Science and Technology, Irbid 22110, Jordan; lneitan@just.edu.jo; 4Department of Family Medicine, Faculty of Medicine, Jordan University of Science and Technology, Irbid 22110, Jordan; 5Department of Internal Medicine, Faculty of Medicine, Jordan University of Science and Technology, Irbid 22110, Jordan; djzahran16@med.just.edu.jo (D.J.Z.); kewant@ccf.org (T.Z.K.); 6Department of Internal Medicine, Cleveland Clinic Foundation, Cleveland, OH 44109, USA

**Keywords:** diffuse large B-cell lymphoma, single nucleotide polymorphism, IL-6, IL-10, Arab population

## Abstract

B-cell lymphomas can be classified as Hodgkin and non-Hodgkin lymphomas. Diffuse large B-cell lymphoma (DLBCL) is the most common non-Hodgkin Lymphoma (NHL). The incidence of NHL is variable and affected by age, gender, racial, and geographic factors. There is strong evidence that the immune-regulatory cytokines have a major role in hematologic malignancies. In this study, we analyzed the relationship between seven single nucleotide polymorphisms (SNPs) in two selected cytokines (IL-6 rs1800795G > C, rs1800796G > C, rs1800797G > A, IL-10 rs1800871G > A, rs1800872G > T, rs1800890A > T, rs1800896T > C) and the risk and overall survival of DLBCL patients in a Jordanian Arab population. One hundred and twenty-five DLBCL patients diagnosed at King Abdullah University Hospital (KAUH) from the period 2013–2018 and 238 matched healthy controls were included in the study. Genomic DNA was extracted from formalin-fixed paraffin-embedded tissues. Genotyping of the genetic polymorphisms was conducted using a sequencing protocol. Our study showed no significant differences in the distribution of all studied polymorphisms of DLBCL between patients and controls. The IL-6 rs1800797 was the only SNP to show significant survival results, DLBCL subjects with the codominant model (GG/AG/AA) genotypes and recessive model (AA genotype in comparison with the combined GG/GA genotype) had worse overall survival (*p* = 0.028 and 0.016, respectively).

## 1. Introduction

Mature B-cell lymphomas are divided into Hodgkin and non-Hodgkin Lymphomas (HL and NHL) [1]. NHL is considered the sixth most common type of cancer and the ninth leading cause of cancer deaths among both males and females [2,3]. The trends of incidence for NHL are variable and show significant age, gender, racial, and geographic differences [4,5,6,7,8]. The surrounding environment, infectious microorganisms, and lifestyle all play an important role in NHL pathogenesis [9,10,11,12]. Diffuse large B-cell lymphoma is the most common NHL [13]. Diffuse large B-cell lymphoma (DLBCL) is a diverse disease and genetically can be classified into two distinct types; the germinal center B-like type, characterized by the expression of germinal center B-cell genes, and the activated B-like type which shows gene expression of activated peripheral blood B cells. Clinically, the germinal center B-like type shows better overall survival [14].

Several studies reveal that the immune-regulatory cytokines have a major role in hematologic malignancies. T-lymphocytes, natural killer cells, macrophages, dendritic cells, and tumor-associated T-lymphocytes are the key components of the tumor inflammatory milieu. Recruitment of inflammatory cells to the site of inflammation is mediated by cytokines. Cytokines are a diverse group of low molecular weight regulatory, mostly soluble, proteins or peptides. Cytokines contribute to cell-to-cell communication, orchestrate the inflammatory and immune responses, act alone, and their secretion is induced by different stimuli. Cytokines are classified according to their molecular structure or function. Structurally, cytokines can be classified into four families: the IL-1 family, the IL-17 family, the tumor necrosis factor (TNF) family, and the four alpha-helix family which includes IL-2, interferon (INF), and IL10. Functionally, cytokines are either part of antibody-mediated (humoral) immunity or cellular-mediated immunity [15].

A number of studies have evaluated the association between specific polymorphisms of IL-6 and IL-10 genes and the risk of non-Hodgkin lymphoma. Fewer studies analyzed the impact of IL-6 and IL-10 polymorphisms on the prognosis of patients with DLBCL [16,17,18].

The serum level of IL-10 affects the overall survival of patients with both HL and NHL. NHL patients with polymorphisms at positions (IL10-3575 T > A rs1800890) and (IL10-1082 A > G rs1800896) show better overall survival (*p* = 0.002) and median overall survival (*p* = 0.05), respectively [19,20]. However, no study has yet described the association between IL-6 and IL-10 gene polymorphisms and the risk and overall survival among an Arab population.

The aim of this study was, therefore, to analyze the relationship between seven, previously studied, single nucleotide polymorphisms (SNPs) in two selected cytokines (IL-6 rs1800795G > C, rs1800796G > C, rs1800797G > A, IL-10 rs1800871G > A, rs1800872G > T, rs1800890A > T, rs1800896T > C) and the risk and overall survival of DLBCL patients in a Jordanian Arab population.

## 2. Results

### 2.1. Demographic and Clinical Data

A total of 363 DNA samples were included in this study, namely, 238 healthy controls and 125 DLBCL patients. Among the patients, 66 (52.8%) were males and 59 (47.2%) were females. Of the control subjects, 92 (38.7%) were males and 146 (61.3%) were females. The age range for the patients was 15–89 years with a mean age of 53.7 years. The mean age for the controls was 43.2 years (6–89). Table 1 summarizes patients’ demographic and clinical data.

### 2.2. Association Between IL-6 and IL-10 Gene Polymorphisms and the Risk of DLBCL

The genotype distributions for all the SNPs in both 125 DLBCL patients and 238 control subjects were in Hardy–Weinberg equilibrium (HWE) and normally distributed with *p*-value > 0.05. The HWE *p*-values for the cases and the controls are shown in Appendix A.

The distribution of both genotypic frequencies and allelic frequencies of IL-6 rs1800795G > C, rs1800796G > C, rs1800797G > A, IL-10 rs1800871G > A, rs1800872G > T, rs1800890A > T, rs1800896T > C SNPs showed no significant differences in the distributions of all studied polymorphisms of IL-6 and IL-10 among DLBCL patients and controls (*p* = 0.87, 0.80, 0.83, 0.76, 0.81, 0.42, and 0.78), respectively. Appendix A.

Additional analyses based on four genetic models (codominant, dominant, recessive, and overdominant) were performed on all the SNPs and the results show no significant association between any of the genetic models and the risk of DLBCL. The results are shown in Appendix A.

### 2.3. Association between IL-6 and IL-10 Gene Polymorphisms and the Survival Rate of DLBCL

The survival analysis for all 125 DLBCL patients was performed using the Kaplan–Meier curve and log-rank test/Wilcoxon (Gehan) statistic. The IL-6 rs1800797 was the only SNP to show significant survival results. In our study, the subjects with the codominant model (GG/AG/AA) genotype and recessive model (AA genotype in comparison with the combined GG/GA genotype) had worse overall survival (OS) (*p* = 0.028 and 0.016), respectively, as shown in Figure 1.

## 3. Discussion

Many cells, including macrophages, B-lymphocytes, T-lymphocytes, mast cells, endothelial cells, fibroblasts, and stromal cells produce cytokines. However, one cytokine can be produced by more than one cell type and the action of one cytokine can be inhibited by another cytokine. The effect of cytokines is pleiotropic, redundant, and synergized; in other words, one cytokine has different effects on different cell types or different cytokines have the same or exaggerated effect [21]. The main function of cytokines is to modulate the inflammatory response by regulating hematopoiesis, innate, and acquired immunity. They control the growth, differentiation, and survival of different cell populations, including inflammatory cells [22]. Cytokines cannot cross the cell lipid bilayered membrane and they exert their function of cell signaling through receptors. Six groups of receptors have been recognized; the immunoglobulin superfamily receptors, the hematopoietin receptor family (Class I cytokine receptor family), the Interferon receptor family (Class II cytokine receptor), the TNF receptor family, the IL-17 receptor family, and the chemokine receptor family [23]. The signaling pathways for cytokines’ receptors include the nuclear factor kappa B (NF-κB), JAK/STAT, and G protein pathways [24,25].

Lymphomagenesis is a complex process that incorporates interaction between the tumor cells and the surrounding stromal cells or extracellular matrix [26]. The inflammatory milieu which incorporates the host inflammatory cells is supposed to exert an immune protective antitumor activity and the pro-inflammatory cytokines should be followed rapidly by the anti-inflammatory cytokines to eradicate inflammation. In cancers, the net balance between immune-mediated antitumor activity and tumor-induced immune suppression is skewed toward the latter [27]. The immune avoidance of tumor cells is a key factor in cancer progression, invasion, and metastasis [28].

Although cytokines may not be involved in early steps of B-cell transformation and lymphoma development, they play an important role in inducing malignant B-cell survival and proliferation and the level of many cytokines is aberrantly elevated in different hematologic malignancies [29,30,31,32]. In addition, cytokine secretion is influenced by the level of gene expression which is determined by host genetic variation or gene polymorphism [33]. The expression and secretion of IL-6 and IL-10 as determined by SNP is associated with risk of developing and or OS of lymphomas [31,34,35].

IL-6 or interferon beta 2 is a 21 kDa, 212 amino acid, monocyte-derived glycoprotein, mapped to chromosome 7 (7p15.3), and assembled as a hexameric complex that includes IL-6, the IL-6 α-receptor (IL-6Rα), and the shared signaling receptor gp130 [36,37]. IL-6 is a pyrogenic, pro-inflammatory, and immune-regulatory cytokine involved in defending the host against infections by stimulating the differentiation of and immunoglobulin synthesis by B-lymphocytes. Although monocyte is the major source for IL-6, it can be produced by many cells including dendritic cells, lymphocytes, neutrophils, mast cells, mesenchymal and stromal cells, glial cells, and tumor cells.

The activation of the innate immune system is a prerequisite and a fundamental step for the initiation of acquired immunity. An essential step in innate immunity response to invading pathogens is to differentiate between self and foreign pathogens, this is done by recognizing biomolecules exclusively found and expressed by the pathogen. These biochemical molecules are proteins, lipids, or sugar signatures known as pathogen-associated molecular patterns (PAMPs) [38]. The immune cells recognize PAMPs by a group of receptors known as pattern recognition receptors (PRRs) which include Toll-like receptors (TLRs), RIG-I-like receptors (RLRs), NOD-like receptors (NLRs), C-type lectin-like receptors (CLRs), and DNA sensors [39].

TLRs are noncatalytic, type 1, transmembrane glycoproteins expressed on macrophages and dendritic cells and recognize conserved structure in the pathogen. In humans, 10 functional TLRs are identified, namely, TLR1–TLR10. The biological responses of each TLR are heterogeneous because the signaling pathways associated with each TLR are not equal [40,41]. The signaling molecules used by the TLRs are either myeloid differentiation primary response gene 88 (MyD88)-dependent or -independent and include MyD88, IL-1R-associated protein kinase, and TNF receptor-activated factor 6. The MyD88-dependent pathway activation leads to the activation of NF-κB transcription factor, which results in the production of pro-inflammatory cytokines [41,42].

Many studies show that IL-6 gene promoter polymorphism (rs1800795G > C) is associated with the risk for developing both neoplastic and non-neoplastic diseases, including Kaposi sarcoma, intracranial hemorrhage, diabetes mellitus, osteoporosis, and Crohn’s disease [43,44,45,46,47]. However, the relationship between IL-6 gene promoter polymorphism and the risk of lymphomas shows inconsistent results. A popular case-control study to assess the relationship between gene polymorphisms and the risk of lymphoma done by the International Lymphoma Epidemiology (InterLymph) consortium in 2006 showed that there is no association between IL-6 promoter polymorphism (174G > C rs1800795) and the risk of NHL [16]. The same conclusion was reached by three more studies; two on Caucasians and one on an Egyptian Arab population which found no association between IL6 rs1800795 (174 G > C) and IL6 rs1800797 (597/598G > A) gene polymorphism and the risk of NHL [48,49,50]. In contrast, Gu et al. found that, in the Han Chinese population, the risk of NHL, in particular, multiple myeloma, is positively associated with IL-6 rs1800795 (GC versus GG: OR = 3.976, 95% CI: 1.400–11.295, *p* = 0.006) and IL-6 rs1800797 (GA versus GG: OR = 3.976, 95% CI: 1.400–11.295, *p* = 0.006) polymorphisms [17]. Consistent with Gu’s result, Ennas et al. found that (IL6-174G > C rs1800795) gene polymorphism is positively associated with the risk of chronic lymphocytic leukemia [18]. The conflicting results may be attributed partially to ethnic and racial factors. In our study on a Jordanian Arab population, the results are consistent with what was found with Caucasians where no significant correlation between the three SNPs (rs1800795G > C, rs1800796G > C, rs1800797G > A) of the IL-6 gene promoter and the risk and OS of DLBCL is present. However, the same thing does not apply to survival findings. In our study, we found that IL-6 rs1800797 SNP shows significant survival results, subjects with the codominant model (GG/AG/AA) genotypes and recessive model (AA genotype in comparison with the combined GG/GA genotype) had worse OS (*p* = 0.028 and 0.016, respectively) than the others.

Interleukin 10 (IL-10), or human cytokine inhibitory factor is a homodimer, pleiotropic anti-inflammatory cytokine that is encoded by the IL-10 gene, which is mapped to chromosome 1q in the junction between 1q31 and 1q32 (1q32.1) [51]. IL-10 is mainly produced by monocytes and acts as anti-inflammatory cytokine by reducing the expression of IL-6 through a receptor complex that inhibits ERK1/2 and NF-κB activation and induces a STAT3 signaling pathway [52]. In addition, IL-10 conducts its anti-apoptotic activity by inducing the expression of human leukocyte antigen G (HLA-G) which inhibits Th1-type cytokines such as TNF-a and IFN-c [53,54].

The expression level of IL-10 messenger RNA is influenced by polymorphisms affecting both the proximal (rs1800872A > C, rs1800890A > T, rs1800896A > G) and the distal (rs1800871T > C) region of IL-10 promoter gene [55]. As a result, IL-10-1082G allele (rs1800896) is associated with high IL-10 production and IL-10-1082A allele (rs1800896) is associated with low IL-10 production [56]. The association between IL-10 gene polymorphisms and IL-10 production and the risk of developing and/or the OS of aggressive NHL is conflicting. In studies done over Chinese populations, the G allele of IL-10 rs1800896 SNP was found to be associated with increased risk of DLBCL [57]. However, other studies show that the low-level IL-10-producing genotypes (−3575A allele of rs1800890 and −1082A allele of rs1800896) show more association with DLBCL [34,58]. The association between the clinical outcome and IL-10 gene polymorphisms in DLBCL also shows different results between different studies and different people. Lech-Maranda et al. reported that DLBCL patients with the presence of G allele (IL-10-1082AG/GG genotype compared to 1082AA genotype) show improved overall survival and a higher rate of complete remission. On the contrary, Mattias Berglund et al. on their study over 168 de novo DLBCL cases found no difference in overall survival between patients with the IL-10-1082AG/GG genotype and patients with the IL-10-1082AA genotype [34,59].

The studies on the relationship between IL-10 rs1800871 C/T and DLBCL risk and overall survival show heterogeneity and inconsistent results among different ethnic groups. Four studies on Caucasians/mixed population done by Lech-Maranda et al., Lan et al., Rothman N et al., and Purdue et al. found no significant association between IL-10 rs1800871 T > C polymorphism and risk of DLBCL in all comparison models [16,34,48,60]. However, Lim et al. in their study over three major races of Malaysian population are the only to show that rs1800871 T > C polymorphism in the recessive model is associated with increased risk of all types of NHL including DLBCL among Malays (*p* = 0.007) and Chinese (*p* = 0.039) but not Indians (*p* = 0.991) [61]. The reasons for the inconsistency in results are uncertain and may be attributed to the variation in mutant alleles frequencies between different populations and ethnic groups (rs1800871, Malay minor allele frequency (MAF) = 0.25; Chinese MAF = 0.27; Indian MAF = 0.37) compared to the Caucasian population in the Rothman et al. study (rs1800871 MAF = 0.75). However, the differences in allelic frequency between ethnicities should not be the only reason responsible for inconsistent results between IL-10 SNPs and susceptibility to DLBCL as heterogeneity is still observed despite consistent allelic frequencies (rs1800871, MAF = 0.32, 0.24, 0.22, and 0.27) for Caucasian populations in Lech-Maranda et al., Lan et al., Purdue et al., and our Jordanian population, respectively, compared to Malaysian populations in the Lim et al. study.

Patient characteristics, treatment options, the dual biological function of IL-10, and the variation in mutant alleles frequencies between different populations and ethnic groups should all be responsible for the inconsistency in the relationship between IL-10 SNPs and DLBCL risk and outcome.

## 4. Material and Methods

### 4.1. Patients and Data Collection

The study population was composed of 125 patients and 238 healthy cancer-free control subjects with similar geographic and ethnic backgrounds to the patients. The 125 cases of DLBCL were retrieved from the archives of King Abdullah University Hospital during the period of 2013–2018. All cases were reviewed by (SK) and one representative section was chosen from each case. All the procedures performed were approved by the ethical committee of Jordan University of Science and Technology (Institutional Review Board (IRB) code number 5/106/2017, dated 8 June 2017) in accordance with the 1964 Declaration of Helsinki and its later amendments. Formal written informed consent was not required with a waiver by the IRB. All control subjects were voluntarily involved and signed written informed consent. Cases’ and controls’ names were coded and blinded and treated confidentially.

### 4.2. DNA Analysis

Genomic DNA was extracted for the DLBCL patients from formalin-fixed paraffin-embedded tissue using a commercially available kit, DNeasy Blood and Tissue Kit (Qiagen Ltd., West Sussex, UK), using the manufacturer’s protocols. Genomic DNA from control-subjects’ blood samples was extracted using the QIAamp^®^ or Promega DNA Mini Kit according to the manufacturer’s instruction. The quality of extracted DNA was examined using agarose gel electrophoresis and ethidium bromide staining. The concentration and purity of extracted DNA were assessed using a NanoDrop 1000^®^ spectrophotometer. The pure DNA samples with their concentrations were sent to the Australian Genome Research Facility (AGRF, Melbourne Node, Melbourne, Australia) for genotyping of seven SNPs in two cytokines’ genes (IL-6 rs1800795G > C, rs1800796G > C, rs1800797G > A, IL-10 rs1800871G > A, rs1800872G > T, rs1800890A > T, rs1800896T > C) in all subjects (patients and controls). The SNPs, SNPs’ position, and primer sequences for IL-6 and IL-10 genes are shown in Table 2. Genotyping with the Sequenom MassARRAY^®^ system (iPLEX GOLD) (Sequenom, San Diego, CA, USA) was performed at the AGRF according to the manufacturer’s recommendations (Sequenom, San Diego, CA, USA). Genotype distributions were compared between patients and controls. Unconditional logistic regression analysis was used to estimate the association between the genotype frequency and the risk of developing DLBCL.

### 4.3. Statistical Analysis

Overall survival (OS) was calculated from the date of diagnosis to the date of death or the last visit for those who were alive at the time of final data collection and analysis. All statistical analyses were performed using IBM SPSS Statistics version 20.0 (IBM Corp., Armonk, NY, USA). The clinical characteristics and response rate of the patients were compared using chi-square tests. The Hardy–Weinberg equilibrium (HWE) test was estimated by a goodness-of-fit χ^2^ test. The Kaplan–Meier method was used to construct survival curves, and the results were compared using a log-rank/Wilcoxon (Gehan) statistic. Multivariate survival was used to control for confounders (age and stage) using Cox regression (Appendix A). The association between polymorphism and the risk for DLBCL was calculated using unconditional logistic regression. The survival curves were displayed using Graph Pad Prism 6 software. All significant variables (*p* < 0.05) were entered into a multivariate model to adjust for possible confounders.

## 5. Conclusions

In this study, we demonstrated that in our Jordanian Arab population the risk of DLBCL is not influenced by any of the polymorphisms affecting either the proximal (rs1800872A > C, rs1800890A > T, rs1800896T > C) or the distal (rs1800871G > A) region of the IL-10 promoter gene (*p* = 0.81, 0.43, 0.78, and 0.76; respectively).

The IL-6 rs1800797 was the only SNP to show significant survival results, DLBCL subjects with the codominant model (GG/AG/AA) genotypes and recessive model (AA genotype in comparison with the combined GG/GA genotype) had worse overall survival (*p* = 0.028 and 0.016, respectively).

## Figures and Tables

**Figure 1 cancers-12-00382-f001:**
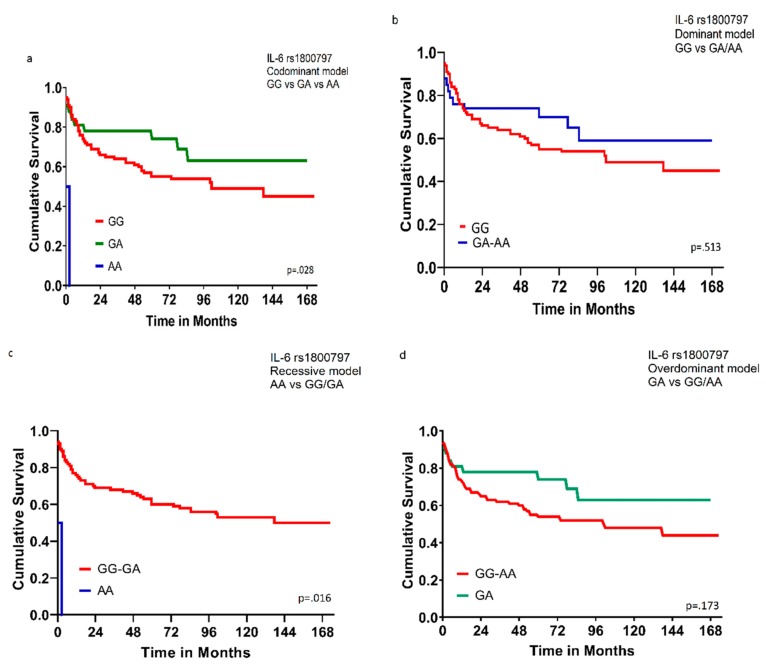
Overall survival for 125 DLBCL patients according to genotype of IL-6 rs1800797. (**a**) Comparison among GG, GA, and AA genotypes; codominant model (*p* = 0.028). (**b**) Comparison between GG genotype and combined GA/AA genotypes; dominant model (*p* = 0.513). (**c**) Comparison between AA genotype and combined GG/GA genotypes; recessive model (*p* = 0.016). (**d**) Comparison between GA genotype and combined GG/AA genotypes; overdominant model (*p* = 0.173). Wilcoxon (Gehan) Statistic *p* values were indicated.

**Table 1 cancers-12-00382-t001:** Demographic and clinical data of 125 diffuse large B-cell lymphoma (DLBCL) patients of Jordanian Arab descent in this study.

**Category**	Value N(%)	
**Demographic Data**	Cases	Controls
**Gender**		
Male	66 (52.8)	92 (38.7)
Female	59 (47.2)	146 (61.3)
**Age in Years ***		
0–14	(0.0)	3 (1.3)
15–19	8 (6.4)	18 (7.6)
20–40	18 (14.4)	89 (37.4)
41–55	31 (24.8)	59 (24.8)
>55	68 (54.4)	69 (28.9)

Mean (Range)	53.7 (1–89)	43.2 (6–89)
Median (IQR)	57 (44–66)	44 (24.2–57)
**Clinical Data**		
**Survival Status**		
Alive	69 (55.2)	−
Dead	56 (44.8)	−
**Survival Months**		−
Median	55	−
**B-Symptoms**		
Yes	15 (14.9)	-
No	86 (85.1)	-
**Ann Arbor Stage at Diagnosis**		
0	2 (1.6)	−
1	25 (20)	−
2	9 (7.2)	−
3	11 (8.8)	−
4	74 (59.2)	−
Unknown	4 (3.2)	−
**Serum LDH**		
Mean (Range)	635 (2–4422)	−
Median (IQR)	423 (194.5–790)	−
**Total Protein**		−
Mean (Range)	58.6 (4–93.3)	−
Median (IQR)	65.8 (57–73)	−
**Serum Albumin**		−
Mean (Range)	35.4 (3–87.8)	−
Median (IQR)	38.4 (33–43)	−
**Total Monocytes**		
Mean (Range)	6.9 (1–22)	−
Median (IQR)	6.3 (4.3–8.5)	−

* Age of controls vs. age at diagnosis in cases. Competition.

**Table 2 cancers-12-00382-t002:** The single nucleotide polymorphisms (SNPs), SNP positions, and primers sequences for IL-6 and IL-10 genes.

SNP-ID	Gene	Chr ^	Bp *	Primer Forward	Primer Reverse
rs1800795	IL-6	7	22727062	ACGTTGGATGAGCCTCAATGACGACCTAAG	ACGTTGGATGGATTGTGCAATGTGACGTCC
rs1800796	IL-6	7	22726627	ACGTTGGATGTCTTCTGTGTTCTGGCTCTC	ACGTTGGATGTGGAGACGCCTTGAAGTAAC
rs1800797	IL-6	7	22726602	ACGTTGGATGTGGAGACGCCTTGAAGTAAC	ACGTTGGATGTCTTCTGTGTTCTGGCTCTC
rs1800871	IL-10	1	206773289	ACGTTGGATGGGTGTACCCTTGTACAGGTG	ACGTTGGATGATGCTAGTCAGGTAGTGCTC
rs1800872	IL-10	1	206773062	ACGTTGGATGAAAGGAGCCTGGAACACATC	ACGTTGGATGTCCTCAAAGTTCCCAAGCAG
rs1800890	IL-10	1	206776020	ACGTTGGATGCAAGCCCAGATGCATAGTAG	ACGTTGGATGCTGATTTCCCAGTACATCCC
rs1800896	IL-10	1	206773552	ACGTTGGATGATTCCATGGAGGCTGGATAG	ACGTTGGATGGACAACACTACTAAGGCTTC

* bp: base pair (Genomic Position). ^ Chr: Chromosome.

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
