# Peer review of "The Impact of IL-6 and IL-10 Gene Polymorphisms in Diffuse Large B-Cell Lymphoma Risk and Overall Survival in an Arab Population: A Case-Control Study"

_cancers, 2020, doi:10.3390/cancers12020382_

Round 1

Reviewer 1 Report

The authors calculate associations of seven known polymorphisms within IL-6 and IL-10 genes with the risk of DLBLC among the Jordanian Arab population. The major conclusion is that none of the polymorphisms studied can be predictive of DLBCL. Only one IL-6 variant has been shown to be correlated with longer survival of patients. The study is well designed and the results are clearly presented but the discussion section requires major improvements:

Major:

(l.138-151) This paragraph is very general and presents information one can find in textbooks of immunology. It should be shortened or deleted (l.165-194) This paragraph is dispensable for the understanding of results presented in this study and should be deleted or shortened (l.231) In a study by Lim et al. doi.org/10.3109/10428194.2014.907895,

      a significant correlation of IL-10 rs1800871 variant with all NHL subtypes among Malaysian and Chinese populations were documented. How can you comment on this? Can these observations be associated with allelic frequency differences between the Jordanian Arab and Malaysian populations? Please include a comment in the discussion paragraph. 

Minor:

1.(l.88) please use capital letters for "Hardy Weinberg" equilibrium

2. (l.255-270) please remove the italic style of this paragraph

Author Response

Response to Reviewer 1 Comments

We appreciate the respected reviewer comments. The following are our point-by- point responses.

Major:

Point 1: (l.138-151) This paragraph is very general and presents information one can find in textbooks of immunology. It should be shortened or deleted 

Response 1: Agree with the respected reviewer. I have, accordingly shortened and modified paragraph (l.138-151).

Point 2: (l.165-194) This paragraph is dispensable for the understanding of results presented in this study and should be deleted or shortened

Response 2: Agree with the respected reviewer. I have, accordingly shortened paragraph (l. 165-194).

Point 3: a significant correlation of IL-10 rs1800871 variant with all NHL subtypes among Malaysian and Chinese populations were documented. How can you comment on this? Can these observations be associated with allelic frequency differences between the Jordanian Arab and Malaysian populations?

Response 3: Thank you for pointing this out. I agree with this comment. Therefore, the following paragraph about the association between IL-10 rs 1800871 gene polymorphism and NHL risk among different ethnic groups has been included in the Discussion (I. 239-254 in revised version and l. 209-224 in edited clean version)

“The studies on the relationship between IL-10 rs1800871 C/T and DLBCL risk and overall survival show heterogeneity and inconsistent results among different ethnic groups. Four studies on Caucasians/mixed population done by Lech-Maranda et al.; Lan et al.; Rothman N et al.; and Purdue et al., found no significant association between IL-10 rs1800871 T>C polymorphism and risk of DLBCL in all comparison models [16, 34, 48, 60]. However; Lim et al., in their study over three major races of Malaysian population are the only to show that rs1800871 T>C polymorphism in the recessive model is associated with increased risk of all types of NHL including DLBCL among Malays (p=0.007) and Chinese (p=0.039) but not Indians (p=0.991) [61]. The reasons for the inconsistency in results are uncertain and may be attributed to the variation in mutant alleles frequencies between different populations and ethnic groups (rs1800871, Malay minor allele frequency [MAF] = 0.25; Chinese MAF = 0.27; Indian MAF = 0.37) compared to the Caucasian population in Rothman et al. study (rs1800871 MAF=0.75). But the differences in allelic frequency between ethnicities should not be the only reason responsible for inconsistent results between IL-10 SNPs and susceptibility to DLBCL as heterogeneity is still observed despite consistent allelic frequencies (rs1800871, MAF = 0.32, 0.24, 0.22, and 0.27) for Caucasian populations in Lech-Maranda et al.; Lan et al.; Purdue et al.; and our Jordanian population, respectively compared to Malaysian populations in Lim et al. study”.

 Minor:

Point 1: (l.88) please use capital letters for "Hardy Weinberg" equilibrium.

Response: Agree with the respected reviewer. Capital letters for Hardy Weinberg used.

Point 2: (l.255-270) please remove the italic style of this paragraph.

Response: Agree with the respected reviewer. Italic style of (l.255-270) was removed.

*Please see the attachment.

Reviewer 2 Report

The manuscript "The impact of IL-6 and IL-10 gene polymorphisms in Diffuse Large B-cell lymphoma risk and overall survival in an Arab population: A case-control study" is a study with several limits.

Several SNPs of IL-6 and IL-10 have been analyzed in a higher number of controls compared to patients. In the control group there are even 3 between 0 and 14 years, but no patients of the same age range are included. the control number is very big, but it was not matched very well with the group of patients. 

In the text there are a lot of general sentences, but no information about the position of the studied SNPs along the genes or about their functionality, and finally their relevance is negligible. 

In my opinion the study is very poor and not acceptable for publication in Cancers.

Author Response

Response to Reviewer 2 Comments

We appreciate the respected reviewer comments. The following are our point-by- point responses.

Point: Several SNPs of IL-6 and IL-10 have been analysed in a higher number of controls compared to patients. In the control group there are even 3 between 0 and 14 years, but no patients of the same age range are included. the control number is very big, but it was not matched very well with the group of patients. In the text there are a lot of general sentences, but no information about the position of the studied SNPs along the genes or about their functionality, and finally their relevance is negligible. 

Response: For our study, the ratio of cases to controls we used is approximately 1 case: 2 controls based on a 1998 article by Susan Lewallen, MD and Paul Courtright, DrPH, entitled “Epidemiology in Practice: Case-Control Studies”. We agree with the respected reviewer in that our case: control groups are not 100% matched, but we think that the efficacy of the study is still statistically valid. Information about the position of studied SNPs along the genes are mentioned in Table 2. The relevance of SNPs are mentioned in the Discussion paragraph, but if the respected reviewer find that we have to elaborate more about the relevance, we are ready to do so. 

Reviewer 3 Report

In case-control study, Al-Khatib et al described the relationship between cytokines gene expressions in Diffuse Large B-cell lymphoma risk and overall survival using SNPs array. This article focused on the expression of two cytokines IL6 and IL10 in the Jordanian Arab population. A Significant correlation was found between IL6 expression and the overall survival. This paper evokes novel ideas such as the role of ethnic and could therefore be regards as deserving of encouragements.  

The role of these cytokines in the clinical outcome of patients as well as the disease have been described previously, as mentioned in the paper, and the heterogeneity of the results did not allow for their  introduction in the clinics and the follow-up of patients.

Major Remarks:

Statistical analysis: it is the core part of the article. The use of multiparametric tests to analyze the relationship between the IL6 expression and overall survival is mandatory. The heterogeneity of used cohort of patients (60% stage IV and 54% more than 55 years) needs to perform a multiparametric test because the stage of the disease is a prognostic biomarker. The introduction of these two factors (stage and age) in the analysis of the results permits a better interpretation. Viral infection: these data are missing in the article The discussion is too long and needs to be rewritten.

Minor remarks:

Table2, 3 and 4 can be put in the supplementary data or else it should (negative results) Figure 1 needs some modifications and the uniformity in the presentation

Author Response

Response to Reviewer 3 Comments

We appreciate the respected reviewer comments. The following are our point-by- point responses.

Major:

Point 1: Statistical analysis: it is the core part of the article. The use of multiparametric tests to analyse the relationship between the IL6 expression and overall survival is mandatory. The heterogeneity of used cohort of patients (60% stage IV and 54% more than 55 years) needs to perform a multiparametric test because the stage of the disease is a prognostic biomarker. The introduction of these two factors (stage and age) in the analysis of the results permits a better interpretation. These data are missing in the article.

Response 1: The authors agree with the respected reviewer. Cox regression analysis was added to the survival analyses accounting for age and stage. (Statistical Analysis (I.304-305 in revised version and l. 269-270 in edited clean version) and Table S4).

Model

OR^ (95% CI)

P-value

Model p-value

Codominant Genotype

0.002

0.000

A/A vs. G/G

1.85 (0.85-4.02)

0.122

A/G vs. G/G

24.25 (4.2-139.8)

0.000

 Age in Years*

0.404

      15-19 vs. 0-14

0.58 (0.08-4.28)

0.59

      20-40 vs. 0-14

0.91 (0.40-2.0)

0.81

      41-55 vs. 0-14

0.46 (0.21-1.0)

0.05

Ann Arbor Stage at Diagnosis

0.407

      1 vs. 0

1.97 (0.23-17.16)

0.540

      2 vs. 0

0.66 (0.32-1.40)

0.280

      3 vs. 0

1.73 (0.62-4.82)

0.297

      4 vs. 0

0.54 (0.16-1.76)

0.303

Dominant Genotype

0.221

       A/G-A/A vs. G/G

1.46 (0.72-2.99)

0.295

 Age in Years*

0.363

      15-19 vs. 0-14

0.53 (0.07-3.95)

0.537

      20-40 vs. 0-14

1.07 (0.48-2.35)

0.874

      41-55 vs. 0-14

0.46 (0.21-1.02)

0.056

Ann Arbor Stage at Diagnosis

0.601

      1 vs. 0

1.97 (0.23-17.16)

0.540

      2 vs. 0

0.66 (0.32-1.40)

0.280

      3 vs. 0

1.73 (0.62-4.82)

0.297

      4 vs. 0

0.54 (0.16-1.76)

0.303

Recessive Genotype

0.000

      G/G-A/G vs. A/A

14.53 (2.87-73.53)

0.001

 Age in Years*

0.391

      15-19 vs. 0-14

0.49 (0.07-3.59)

0.48

      20-40 vs. 0-14

0.91 (0.4-2.05)

0.814

      41-55 vs. 0-14

0.46 (0.21-1.01)

0.053

Ann Arbor Stage at Diagnosis

0.391

       1 vs. 0

1.25 (0.16-9.95)

0.833

       2 vs. 0

0.71 (0.34-1.49)

0.363

       3 vs. 0

2.02 (0.73-5.57)

0.173

       4 vs. 0

0.54 (0.17-1.79)

0.317

Overdominant Genotype

0.101

0.145

       G/G-A/G vs. A/A

0.52 (0.24-1.13)

0.101

 Age in Years*

0.371

      15-19 vs. 0-14

0.57 (0.08-4.25)

0.586

      20-40 vs. 0-14

1.06 (0.48-2.32)

0.892

      41-55 vs. 0-14

0.46 (0.21-1.01)

0.054

Ann Arbor Stage at Diagnosis

0.585

      1 vs. 0

1.77 (0.21-15.28)

0.604

      2 vs. 0

0.74 (0.36-1.54)

0.423

      3 vs. 0

1.58 (0.57-4.37)

0.382

      4 vs. 0

0.54 (0.16-1.78)

0.310

Table S4: Multivariate Survival analyses using Cox Regression Model for SNP 1800797 and controlling for Age and Stage.

Point 2: The discussion is too long and needs to be rewritten.

Response 2: Agree with the respected reviewer. The discussion paragraph been modified and shortened.

Minor:

Point 1: Table2, 3 and 4 can be put in the supplementary data.

Response 1: Agree with the respected reviewer. Tables 2, 3, and 4 been relocated to supplementary data and appear as: Table S1, Table S2, and Table S3; respectively.  

Point 2: Figure 1 needs some modifications and the uniformity in the presentation.

Response 2: Agree with the respected reviewer. Figure 1 is modified, panels a, b, c, and d are arranged as square rather than L-shape and made more uniformed.

Round 2

Reviewer 2 Report

Author's answer: Information about the position of studied SNPs along the genes are mentioned in Table 2. The relevance of SNPs are mentioned in the Discussion paragraph, but if the respected reviewer find that we have to elaborate more about the relevance, we are ready to do so. 

In my opinion when SNP position is requested to indicate the precise position (base pair number) is not very informative. What that can be important is to know in wich region the SNP is: coding region? 3'-UTR? promoter?....and so on.

When some SNPs are analyzed it is important to understand and describe if they allow aminoacid changes or not, the formation of a stop codon or not, or if nothing changes. This can be describe in the table with a new column.

Reviewer 3 Report

The authors responded clearly to all the suggestions. I think the paper deserves to be published now